# Association of relationship satisfaction with blood pressure: a cross-sectional study of older adults in rural Burkina Faso

Felicitas Maria Jaspert [ID],[1,2] Guy Harling [ID],[3,4,5,6] Ali Sie,[7] Mamadou Bountogo,[7] Till Bärnighausen,[4,8,9] Beate Ditzen,[1,9,10] Melanie Sandy Fischer[1,11]

For numbered affiliations see end of article.

**Correspondence to**
Dr Melanie Sandy Fischer; melanie.fischer@uni-marburg.de

## ABSTRACT

**Objectives** The objective of this study is to examine the association between relationship satisfaction and blood pressure (BP) in a low-income setting and to screen for gender moderation in this context. Research conducted in high-income settings suggests that relationship satisfaction is associated with better physical and mental health outcomes.

**Design** A cross-sectional study design was employed, using both questionnaire and physical measurement data. Multiple linear regression models were calculated for systolic and diastolic BP and adjusted for age, gender, demographics/socioeconomics and other health-related variables. Gender moderation was tested using interaction terms in multivariable analyses.

**Setting** A household survey was conducted in 2018 in rural northwestern Burkina Faso.

**Participants** Final analysis included 2114 participants aged over 40 who were not pregnant, reported being in a partnership and had valid BP readings.

**Main outcome measures** Systolic and diastolic BP levels.

**Results** A significant positive association existed between relationship satisfaction (Couples Satisfaction Index-4 score) and systolic BP (B=0.23, 95% CI (0.02 to 0.45), p=0.03) when controlling for demographics/socioeconomics. Nevertheless, this relationship lost statistical significance when additional adjustments were made for health-related variables (B=0.21, 95% CI (−0.01 to 0.42), p=0.06). There was no significant association of relationship satisfaction and diastolic BP and no evidence of gender moderation.

**Conclusion** In contrast to many higher-income settings, we found a positive association between relationship satisfaction and systolic BP in very low-income rural Burkina Faso. Our results add to the evidence regarding the contextual nature of the association between relationship satisfaction and health, as high relationship satisfaction may not act as a health promotor in this socioeconomic context.

## STRENGTHS AND LIMITATIONS OF THIS STUDY

⇒ Population-based design: The study's large, representative sample of older adults from rural Burkina Faso enhances the generalisability of the findings to similar sociodemographic settings in sub-Saharan Africa.

⇒ Objective measurements: The use of standardised questionnaires and physical measurements allows for objective quantification of health parameters, ensuring comparability with other studies.

⇒ High Reliability of the Couples Satisfaction Index-4 (CSI-4): The CSI-4 demonstrated high reliability in the sample, supporting the robustness of the relationship satisfaction measure used.

⇒ Measurement constraints: The study's reliance on a brief, unidimensional measure of relationship satisfaction (CSI-4) and the use of clinical blood pressure measurements, rather than ambulatory methods, may have limited the ability to capture nuanced variations in relationship satisfaction and its potential associations with blood pressure.

⇒ Cross-sectional design: The cross-sectional nature of the study limits the ability to infer causation, highlighting the need for longitudinal research to better understand the causal pathways between relationship satisfaction and blood pressure in this context.

## INTRODUCTION

Between 2019 and 2050, sub-Saharan Africa will account for more than half of the world population growth with a predicted population rise of one billion people.[1] Meanwhile, life expectancy in sub-Saharan Africa has steadily improved from 49.4 years in 1990 to 62.9 years in 2019 and is projected to reach 68.5 years by 2050.[1] Based on population growth and population ageing, the proportion and absolute number of older people in sub-Saharan Africa is expected to increase. As demographics change, non-communicable diseases, including cardiovascular diseases (CVDs), are on the rise. This is reflected in the 67% increase in total disability-adjusted life years associated with non-communicable diseases in sub-Saharan Africa between 1990 and 2017.[2] In 2017, the age-standardised disability-adjusted life year rate due to non-communicable diseases was almost as high as

the one due to communicable, maternal, neonatal and nutritional diseases, the main burden of disease in sub-Saharan Africa to date.[2] Among the non-communicable diseases causing disability-adjusted life years, CVDs account for the largest share from age 40 onwards, making them particularly impactful on the older population in sub-Saharan Africa.[2] The systemic role of CVD is emphasised by the fact that in most low-income and middle-income countries, the main cause of the decline in non-communicable disease mortality is due to the decline in deaths from CVD.[3] It is, therefore, especially important to target the main risk factors for CVD to address the increasing public health burden of non-communicable diseases in sub-Saharan Africa.

Social relationships are associated with the development and progression of CVD[4 5] and are comparable in effect size to established health risk factors.[6] Epidemiological studies have shown that lower social support is associated with a higher mortality rate of affected individuals, especially due to CVDs.[5 7] However, to date, studies examining the impact of social support and relationships on health have been conducted mainly in high-income Western countries.

When considering sources of social support, close relationships such as marriage and romantic partnership play a key role. Comparing the effects of different social relationships on well-being, the quality of marriage is more strongly associated with health than the quality of other family and friendship relationships,[8] making spousal relationships a particularly important target in health promotion. Although married people generally exhibit better health than unmarried people, as demonstrated in a meta-analysis by Robles et al,[9] the mere fact of being in a partnership or marriage does not in itself necessarily bring benefits, but rather a satisfaction and support within such a relationship.[10]

Relationship satisfaction is the most important construct studied to represent relationship quality,[11] as well as a core construct in basic relationship research.[12] Relationship quality is measured almost exclusively by means of self-reports, thematically built on two main approaches: First, on interpersonal aspects of specific couple interactions such as communication and conflict, and second, on intrapersonal aspects such as a global, subjective evaluation of the relationship. Relationship satisfaction tends to be assigned to the second approach, although items in established measurement scales often overlap with both.[11] In Funk and Rogge's[12] evaluation of measurement scales assessing relationship satisfaction, self-reports focusing on the global assessment of partnership were found to have more information, higher precision and greater explanatory power with respect to group differences, arguing for more focused measurements of relationship satisfaction using intrapersonal aspects. Since satisfaction measures based on global ratings of relationship satisfaction are less dependent on culturally determined interaction processes (such as communication norms) than measures based on ratings of specific

relationship behaviours,[13 14] relationship satisfaction is a readily applicable measure for assessing couple relationships in a non-Western context.

In Africa, very little research exists specifically addressing relationship satisfaction or its impact.[15–17] Previous research in Western countries demonstrates a significant association between high relationship satisfaction and better cardiovascular health, including lower blood pressure (BP), reduced risk of coronary heart disease, improved heart rate variability, as well as decreased incidence of myocardial infarction.[9 18 19] Nevertheless, research specifically addressing the association of relationship satisfaction with BP remains scarce, with studies reporting mostly small, positive effect sizes, along with significant heterogeneity across studies.[9] To date, several psychophysiological pathways have been proposed to explain the health-promoting effects of high relationship satisfaction, including improved long-term health behaviours, effects on the endocrine and immune systems and chronic stress-buffering effects.[9 18 20–22] Central to psychosocial research on CVD is the 'reactivity hypothesis', which posits that stress triggers cardiovascular reactivity, such as elevated BP and heart rate, as well as increases in stress hormone levels and the promotion of inflammation, all of which contribute to the development and progression of CVD.[18 22] Reviews of the literature show that marital quality significantly affects cardiovascular reactivity, with negative aspects, such as hostility, leading to increased BP and heart rate in marital conflict and positive aspects, such as support, buffering stress reactivity.[9 23] However, the association of relationship satisfaction with cardiovascular health is complex. Particularly in high-quality relationships, stress can be transmitted between partners if one partner has health problems. This potential stress-spillover effect might influence the link between relationship quality and health.[23] As far as we know, our study marks the first investigation into the association between relationship satisfaction and cardiovascular health outcomes in African countries. Concurrent research conducted by Kurniawan et al[24] within the same survey explored the association between relationship satisfaction and metabolic outcomes, expanding on our inquiry. This represents an initial step in addressing the need for targeted relationship research in this socioeconomic and cultural context.

To determine the association of relationship satisfaction with CVD, surrogate endpoints that serve as biomarkers for CVD risk can be used as health endpoints. High BP, a reliable surrogate endpoint and the main risk factor for CVD, is most strongly evidenced as a causative factor and highly prevalent in exposure.[25 26] With the shift in world's highest BP values from high-income to low-income countries, sub-Saharan Africa has become one of the regions with the highest mean BP levels and increasing prevalence of elevated BP.[27] Systolic BP (SBP) ranked first among modifiable risk factors for CVD burden in 2019[28] and given the high prevalence and impact of elevated BP as the most common risk factor for developing

CVD in sub-Saharan Africa,[29] elevated BP is particularly important as a key factor for CVD preventability. In sub-Saharan Africa, many established risk factors for elevated BP such as age, high body mass index (BMI), a sedentary lifestyle, high salt intake, alcohol consumption and smoking have already been confirmed and studied,[29 30] while data on the influence of relationship satisfaction on BP remain scarce or even non-existent. Research on composite indices distinguishing cardiovascular health levels and high normal BP in older adults is just beginning to emerge,[31] and considering that relationship satisfaction is a potentially modifiable risk factor—such as through relationship education programmes[32]—a more refined understanding of these associations could significantly inform targeted interventions.

Apart from the lack of research on the health impacts of relationship satisfaction in sub-Saharan Africa in general, there is an ongoing need to clarify possible moderating effects. Literature on health effects of relationship quality shows evidence of moderation by gender. This moderation is particularly evidenced when cardiovascular surrogate endpoints are used as outcomes and tend to indicate a stronger association of relationship quality and cardiovascular health for women than for men.[9] Nevertheless, most studies conducted separate analyses for gender and did not test directly whether effects were significantly different for men and women and found mostly small, non-significant differences, where it can be assumed that the sample size used was not sufficient to detect such small effect sizes.

### Current study

The current study examined the association between relationship satisfaction and BP, a surrogate endpoint to future cardiovascular risk, in older people of Burkina Faso, a low-income country in Western Sub-Saharan Africa, and the moderating role of gender in this context using a large, representative sample. This is the first investigation of relationship satisfaction and cardiovascular health in sub-Saharan Africa focusing on the potential health impact of relationship satisfaction on clinically important, biological and modifiable CVD risk factors.

Marriage practices in Burkina Faso differ significantly from Western norms. Marriage is nearly universal, with women often entering unions before age 18 through arranged marriages, frequently involving significant age disparities. Polygamous marriages are widespread, especially in rural areas, and are commonly associated with lower levels of education.[33 34] While such marriages are generally perceived positively by most Burkinabe women[35] and can offer benefits like emotional support among co-wives, shared child-rearing responsibilities and economic advantages,[36] research indicates that women in polygamous marriages exhibit significantly higher rates of mental health issues and report greater difficulties in family functioning, marital relationships and overall life satisfaction, with senior wives being more affected than junior wives.[33 37 38] The frequency of divorces initiated by women is increasing, alongside a rise in remarriages.[35 39] When examining relationship satisfaction in this context, it is crucial to consider the unique aspects of relationship formation and marital expectations: While Western marriages often emphasise passionate love and intimacy, Burkinabe unions may prioritise family obligations and economic considerations.[39] Therefore, in arranged marriages, relationship satisfaction may be more closely tied to fulfilling familial duties and achieving social stability rather than personal emotional fulfilment. Nevertheless, an individual's global assessment of their relationship satisfaction, even if based on factors specific to this cultural context, may still play an important role in terms of health outcomes.

Additionally, Burkina Faso faces a high prevalence of CVDs, with 2 CVDs ranking among the top 10 causes of death in 2019, and high BP being the fourth major risk factor for mortality and disability.[40] This reinforces the relevance of our research question in this specific country and unique cultural and health context, potentially suggesting relevant societal and health policy directions.

Based on previous research from Western countries, we expected (a) that higher relationship satisfaction would have a protective and thus negative association with SBP and diastolic BP (DBP) levels and (b) that gender would moderate the relationship, with women showing a stronger association than men.

## METHODS
### Study area and population

Burkina Faso, a land-locked country in sub-Saharan Africa, is ranked as one of the lowest-income countries in the world. In 2016, Burkina Faso was listed as the seventh last country in terms of the Human Development Index with 64.8% of the population living in severe multidimensional poverty.[41] The sample used for the study was drawn from the population of the Nouna Health and Demographic Surveillance System (HDSS), located in the Kossi Province, western Burkina Faso. The sample is representative of the population of Burkina Faso with appropriate caution.[42] The HDSS includes the semi-urban city of Nouna and 58 surrounding villages, a study area with a total population of ~107000 people living in around 15000 households according to the 2015 census. The survey was conducted by the Centre de Recherche en Santé de Nouna (CRSN), the national health research centre. The mostly rural study population relies primarily on farming and cattle keeping. Major ethnic groups that make up the multiethnic health district are Dafing, Bwaba, Mossi, Peulh and Samo. Islam and the Roman Catholic Church are the most prominent religions represented. Besides the official language French, there are several local dialects, with Dioula being the most widely spoken. Most children receive hardly any education, averaging 1 year of schooling for women and 2 years for men.[41] In 2018, less than half of adults (≥15 years) could both read and write.[42 43] Marriage is the most common relationship

status for adults in Burkina Faso. By the age of 25, 95% of women are married. 58% of the population in marital unions live monogamously, with polygamy also being widespread, concerning almost 42% of married women.[44]

## Procedure

The study used cross-sectional data collected as part of the CRSN Heidelberg Aging Study, a household survey conducted in the Nouna HDSS between May and July 2018. The primary objective of the CRSN Heidelberg Aging Study was to conduct a quantitative survey to assess the health status of older adults (≥40 years) in rural Burkina Faso. The age-eligible population consisted of approximately 18 000 individuals from the 2015 HDSS census. With the target of 3000 responses and the assumption of 25% loss due to mortality, mobility or non-response, 3998 individuals were selected using a stratified two-stage cluster random sampling approach, explained in more detail elsewhere.[45] Selected individuals were asked for written informed consent, with a witness in case of illiteracy. Data collection was carried out by trained local fieldworkers in the form of a structured questionnaire at the respondents' place of residence. It consisted of an interview on sociodemographic characteristics, physical and mental health, and healthcare utilisation, conducted either in French or translated into Dioula. The interviews were individual and either public or private, as chosen by the interviewees. Furthermore, physical measurements and a blood sample were taken by certified research staff. A blank copy of the questionnaire, including relevant sections used in this study, as well as the French version of the Couple Satisfaction Index-4 (CSI-4), is provided as online supplemental material.

## Measures

### Relationship satisfaction

Considering the sociocultural background as well as the size of the sample, an applicable, brief, reliable and valid measure of relationship satisfaction was aimed for. Given these requirements, the CSI-4 was selected.[12] The CSI-4 is a psychometrically optimised four-item self-report scale designed to assess relationship satisfaction in married or cohabiting couples. Items assess relationship satisfaction at a global level (eg, 'In general, how satisfied are you with your relationship?'). The items are typically rated on a 7-point Likert scale for the first item (0=extremely unhappy; 6=perfect) and on a 6-point Likert scale for the other three items (0=not at all; 5=perfect). The CSI-4 was translated to French for the purpose of the CRSN Heidelberg Aging Study survey and, like other self-report measures in the survey, was administered as an interview by research staff. In order to optimise the measure for administration in the CRSN Heidelberg Aging Study, all items were rated on a 6-point Likert scale, thus making the highest score of the first item 5=extremely happy. Item scores are summed, with higher total scores reflecting higher levels of overall relationship satisfaction. The CSI-4 total score ranges from 0 to 20 (usually: 0 – 21)

and is used as a continuous score. Internal consistency (Cronbach's alpha) in the current sample was α=0.84. A CSI-4 score of 13.5 or lower has been established as a cut-off to indicate relationship distress.[12] Compared with traditional relationship satisfaction measurements, such as the Marital Adjustment Test and the Dyadic Adjustment Scale, which have already been applied in Burkina Faso,[16] the CSI-4, although much shorter, provides a large amount of information, demonstrating a high degree of precision and power.[12]

### Blood pressure

Following the recommendations of the WHO STEPS Surveillance Manual,[46] BP—SBP and DBP—were measured a total of three times in the left arm after at least 15 min of rest in a seated position. Measurements were performed using a portable BP Monitor (Omron Series 7) at 5 min intervals between the second and third reading. The mean systolic and diastolic values of the last two measurements were calculated to minimise random errors and ensure a more accurate estimate of BP. The mean values were included in the analyses as continuous variables. Hypertension was defined using the cut-off values set by current guidelines from the American College of Cardiology/American Heart Association (SBP ≥130 mmHg or DBP ≥80 mmHg).[47]

### Covariates

This study accounted for age and gender, relevant demographic and socioeconomic variables as well as health-related variables. The selection of covariates was based on their established relevance in epidemiological studies and their known influence on both relationship satisfaction and BP, aiming to control for potential confounding factors. Age was included as a continuous variable, and gender as a binary variable with men as the reference group. Demographic and socioeconomic variables included wealth quintiles, defined using a wealth index based on important housing features and household ownership of durable assets,[48] categorised from quintile 1 (poorest=reference group) to 5 (richest). Education was coded as a binary variable where 0 represented those who did not complete primary school (less than primary=reference group) and 1 represented those who completed primary school or attained higher levels of education (some=primary completed or more). In the context of Burkina Faso, primary education consists of 6 years of formal schooling, and completing this level is considered a significant achievement in education attainment, particularly in rural areas where access to education is limited.[49] Because the number of participants who completed any additional schooling past primary school was extremely small (4.1%), we combined education levels into these two categories. Ethnicity was included with the five main ethnic groups coded separately and others as the reference group. Health-related variables included BMI, depression measured using the Patient Health Questionnaire for Depression-9 and sitting hours during a usual

week (Sit-h/week), an indicator of physical activity, all used as continuous variables.

## Patient and public involvement

Patients and/or the public were not involved in the design, or conduct, or reporting, or dissemination plans of this research.

## Statistical analyses

To account for potential confounding effects of covariates that might influence both relationship satisfaction and BP values, Pearson's correlation coefficients were calculated for all pairs of continuous variables to assess linear relationships. For categorical variables, mean relationship satisfaction and BP values were computed for each category. Differences between binary categories were assessed using t-tests, while differences across multiple categories were evaluated using one-way analysis of variance (ANOVA), providing insight into the statistical significance of group differences.

Multiple linear regression models were then applied to test the association between relationship satisfaction and SBP as well as DBP. All models were adjusted for age and gender. The effect of relationship satisfaction on BP was assessed when controlling for demographic and socioeconomic characteristics (model 1) and when additionally controlling for relevant health-related variables (model 2). The individual conditions for the application of multiple linear regression models, such as the absence of multicollinearity or normal distribution of the residuals, were checked.

To investigate whether the association between relationship satisfaction and BP was moderated by gender, an interaction term (gender×CSI-4) was created, and multivariable linear regression models were run for SBP and DBP with all the above covariates and the interaction term. All analyses were performed using R Studio.[50]

**Table 1** Sociodemographic characteristics of included participants (N=2114)

| | Total (%) | Female (%) | Male (%) |
|---|---|---|---|
| | N=2114 (100) | N=826 (39.07) | N=1288 (60.93) |
| Age mean (years) | 51.7 | 51 | 52.1 |
| Marital status | | | |
| Never married | 17 (0.80) | 7 (0.85) | 10 (0.78) |
| Currently married | 1952 (92.34) | 759 (91.89) | 1193 (92.62) |
| Cohabiting | 124 (5.87) | 46 (5.57) | 78 (6.06) |
| Seperated, divorced or widowed* | 21 (0.99) | 14 (1.70) | 7 (0.55) |
| Wealth quintiles | | | |
| Poorest quintile | 332 (15.70) | 140 (16.95) | 192 (14.91) |
| Second poorest quintile | 442 (20.91) | 176 (21.31) | 266 (20.65) |
| Middle quintile | 451 (21.33) | 168 (20.34) | 283 (21.97) |
| Second richest quintile | 466 (22.04) | 185 (22.40) | 281 (21.82) |
| Richest quintile | 423 (20.01) | 157 (19.01) | 266 (20.65) |
| Ethnicity | | | |
| Dafin | 834 (39.45) | 306 (37.05) | 528 (40.99) |
| Bwama | 623 (29.47) | 263 (31.84) | 360 (27.95) |
| Mossi | 282 (13.34) | 105 (12.17) | 177 (13.74) |
| Peulh | 186 (8.80) | 80 (9.69) | 106 (8.23) |
| Samo | 150 (7.10) | 57 (6.90) | 93 (7.22) |
| Other | 39 (1.84) | 15 (1.82) | 24 (1.86) |
| Language | | | |
| French | 73 (3.45) | 5 (0.61) | 68 (5.28) |
| Dioula | 1857 (87.84) | 742 (89.83) | 1115 (86.57) |
| Other | 184 (8.71) | 79 (9.56) | 105 (8.15) |
| Education | | | |
| Less than primary | 1958 (92.62) | 797 (96.49) | 1161 (90.14) |
| Primary school or more | 156 (7.38) | 29 (3.51) | 127 (9.86) |

*Categories were combined in order to avoid reporting on cells with <3 participants.

Table 2   Descriptive characteristics of included participants (N=2114): means and SD for the total sample, women and men and t-tests comparing gender group differences

| | Total | | Women | | Men | | T-tests |
|---|---|---|---|---|---|---|---|
| | N=2114 | | N=826 | | N=1288 | | |
| | **Mean** | **SD** | **Mean** | **SD** | **Mean** | **SD** | **t** |
| CSI-4 | 12.2 | 3.6 | 11.6 | 3.7 | 12.5 | 3.5 | 5.43*** |
| SBP* | 125.22 | 18.22 | 123.01 | 19.20 | 126.63 | 17.43 | 4.48*** |
| DBP | 80.54 | 10.98 | 79.83 | 10.99 | 80.99 | 10.96 | 2.38* |
| BMI† | 22.10 | 3.90 | 22.56 | 4.53 | 21.80 | 3.41 | −4.41*** |
| PHQ-9 | 4 | 3.4 | 4.3 | 3.3 | 3.7 | 3.4 | −4.02*** |
| Sit-h/week | 24.1 | 12.6 | 24 | 12.2 | 24.2 | 12.8 | 0.53 |

| | Total | | Women | | Men | |
|---|---|---|---|---|---|---|
| | **N** | **%** | **N** | **%** | **N** | **%** |
| Hypertensive‡ | 1140 | 53.93 | 418 | 50.61 | 722 | 56.06 |
| Stage I§ | 637 | 30.13 | 229 | 27.72 | 408 | 31.68 |
| Stage II¶ | 503 | 23.79 | 189 | 22.88 | 314 | 24.38 |

*p<0.05, ***p<0.001.
*BP (mmHg).
†BMI [kg/m$^2$]
‡SBP ≥130 mmHg or DBP ≥80 mmHg
§SBP 130-139 mmHg or DBP 80-89 mmHg
¶SBP ≥140 mmHg or DBP ≥ 90 mmHg
BMI, body mass index; CSI-4, Couple Satisfaction Index-4; DBP, diastolic blood pressure; PHQ-9, Patient Health Questionnaire; SBP, systolic blood pressure; Sit-h/week, Sitting hours per usual week.

## RESULTS

Of the 3998 people sampled, 3026 were willing and able to give informed consent to participate and provided baseline information on sociodemographic variables. The subsample used in our analyses included only respondents who were not pregnant at the time of the survey (N=2969), as pregnancy is known to be associated with significant physiological changes including BP levels.[51] In the subsample, 2254 study participants reported being in a partnership and provided information on relationship satisfaction. Of these, averaged SBP and DBP readings were collected from 2222 individuals.

101 participants stated having taken BP medication in the last 2 weeks and were, therefore, not included in the analyses. Six participants with missing values for BMI and Sit-h/week, as well as one outlier with an extreme value for Sit-h/week (168 hours=7×24 hours) were removed from the sample, giving a total subsample size of N=2114 participants. Summary statistics were compiled for the total subsample and for women and men individually and can be found in table 1. To protect participant confidentiality and prevent identification of individuals in groups with small numbers, the marital statuses of separated, divorced and widowed were combined into

Table 3   Pearson correlation matrix of continuous variables

| Variables | CSI-4 | SBP | DBP | Age | BMI | PHQ-9 | Sit-h/week |
|---|---|---|---|---|---|---|---|
| CSI-4 | 1 | | | | | | |
| SBP | 0.05* | 1 | | | | | |
| DBP | 0.04. | 0.78*** | 1 | | | | |
| Age | −0.04. | 0.21*** | 0.02 | 1 | | | |
| BMI | 0.07** | 0.12*** | 0.16*** | −0.17*** | 1 | | |
| PHQ-9 | −0.15*** | 0.06** | 0.00 | 0.29*** | −0.11*** | 1 | |
| Sit-h/week | 0.00 | 0.09*** | 0.04* | 0.23*** | −0.01 | 0.17*** | 1 |

.p < .1 and *p < .05 and **p < .01 and ***p < .001
BMI, body mass index; CSI-4, Couple Satisfaction Index-4; DBP, diastolic blood pressure; PHQ-9, Patient Health Questionnaire for Depression; SBP, systolic blood pressure; Sit-h/week, Sitting hours per usual week.

**Table 4** Comparison of mean relationship satisfaction, systolic blood pressure and diastolic blood pressure by categorical variables with t-test and ANOVA results

| Variable* | Category | Relationship satisfaction | | Systolic blood pressure | | Diastolic blood pressure | |
|---|---|---|---|---|---|---|---|
| | | Mean | T | Mean | T | Mean | T |
| Education | Less than primary | 12.1 | −2.34* | 125.06 | −1.42 | 80.31 | −3.36*** |
| | Some | 12.8 | | 127.22 | | 83.38 | |
| | | **Mean** | **F-value** | **Mean** | **F-value** | **Mean** | **F-value** |
| Ethnicity | Dafin | 12.4 | 5.21*** | 125.45 | 2.46* | 80.59 | 1.41 |
| | Bwama | 11.7 | | 125.96 | | 80.70 | |
| | Mossi | 12 | | 124.06 | | 80.25 | |
| | Peulh | 12.9 | | 121.49 | | 79.22 | |
| | Samo | 12.1 | | 127.19 | | 80.82 | |
| | Other | 12.5 | | 127.08 | | 84.03 | |
| Wealth quintiles | Poorest quintile | 11.4 | 10.78*** | 124.88 | 1.30 | 80.00 | 4.49** |
| | Second poorest quintile | 11.9 | | 123.78 | | 79.34 | |
| | Middle quintile | 11.9 | | 125.05 | | 80.38 | |
| | Second richest quintile | 12.5 | | 126.18 | | 80.55 | |
| | Richest quintile | 12.9 | | 126.10 | | 82.36 | |

*p < .05, **p < .01 and ***p < .001
*Gender differences are reported in table 2.
ANOVA, analysis of variance.

a single category. Descriptive statistics for all study variables are reported in table 2.

The results of the correlation matrix can be found in table 3, and the statistical comparisons across categories are detailed in table 4. Relationship satisfaction and SBP were significantly but weakly positively correlated, r(2112)=0.05, p=0.024, while relationship satisfaction and DBP were not significantly correlated, r(2112)=0.04, p=0.056. Given the significant positive association between BMI, relationship satisfaction and BP, as well as the higher levels of relationship satisfaction overserved in wealthier quintiles, a post hoc ANOVA was performed to examine whether higher wealth is associated with a higher BMI, followed by a Tukey-Honestly Significant Difference (HSD) test to identify specific differences between wealth quintiles. The analysis revealed significant differences in BMI between wealth quintiles, with individuals in higher wealth quintiles exhibiting significantly higher BMI scores compared with those in lower quintiles (see online supplemental material for mean BMI comparisons by wealth quintiles with ANOVA and Tukey-HSD results). The results of the multiple linear regression analyses are presented in tables 5 and 6 (SBP) and (DBP).

### Systolic blood pressure

Relationship satisfaction was significantly and positively (B=0.23; 95% CI (0.02 to 0.45); p=0.03) associated with SBP in model 1 which controlled for demographic and socioeconomic variables, indicating that individuals who

were more satisfied in their relationship had slightly higher SBP values. When relevant health parameters were added to the model (model 2), the association between relationship satisfaction and SBP was no longer significant (B=0.21; 95% CI (−0.01 to 0.42); p=0.06). In both models for SBP, age and gender were significant covariates. Age was positively associated with SBP in models 1 and 2, and women had lower SBP compared with men. In model 2, BMI was also significantly and positively associated with SBP. The adjusted $R^2$ was 0.06 for model 1 and 0.08 for model 2, indicating that the predictors used in each model jointly explained about 6% and 8% of the variance in SBP.

### Diastolic blood pressure

No significant association was found between relationship satisfaction and DBP, neither in model 1 (B=0.09; 95% CI (−0.04 to 0.22); p=0.18), which controlled for demographic and socioeconomic variables, nor in model 2 (B=0.07; 95% CI (−0.06 to 0.21); p=0.28), when relevant health parameters were added. In model 1, the wealthiest, represented by wealth quintile 5, had significantly higher DBP compared with the poorest, as did the educated compared with the uneducated. In model 2, significant differences in DBP existed between genders, with women having lower DBP compared with the reference group of men. Also, BMI was significantly and positively associated with DBP. The adjusted $R^2$ was 0.01 for model 1 and 0.03 for model 2.

Table 5  Results of regression models estimating the association between relationship satisfaction and systolic blood pressure, controlling for demographic and socioeconomic characteristics (model 1) and additionally relevant health-related variables (model 2)

| | Model 1 | | | Model 2 | | |
|---|---|---|---|---|---|---|
| | Estimate (B) | 95% CI | P value | Estimate (B) | 95% CI | P value |
| CSI-4 | 0.23 | (0.02; 0.45) | 0.03 | 0.21 | (−0.01; 0.42) | 0.06 |
| Age (years) | 0.42 | (0.34; 0.50) | <0.001 | 0.43 | (0.34; 0.51) | <0.001 |
| Female (vs male) | −2.83 | (−4.41; −1.26) | <0.001 | −3.60 | (−5.17; −2.03) | <0.001 |
| Household wealth (vs poorest quintile) | | | | | | |
|   Second poorest quintile | 0.22 | (−2.33; 2.77) | 0.87 | −0.09 | (−2.61; 2.43) | 0.95 |
|   Middle quintile | 1.09 | (−1.47; 3.65) | 0.40 | 0.27 | (−2.28; 2.82) | 0.84 |
|   Second richest quintile | 2.36 | (−0.22; 4.94) | 0.07 | 1.55 | (−1.02; 4.11) | 0.24 |
|   Richest quintile | 2.03 | (−0.67; 4.73) | 0.14 | −0.29 | (−3.04; 2.46) | 0.83 |
| Education (vs no education) | 1.76 | (−1.28; 4.81) | 0.26 | 1.02 | (−2.00; 4.04) | 0.51 |
| Ethnicity (vs other ethnicities) | | | | | | |
|   Bwaba | −1.34 | (−7.13; 4.46) | 0.65 | −0.87 | (−6.60; 4.86) | 0.77 |
|   Dafin | −2.95 | (−8.69; 2.80) | 0.31 | −2.36 | (−8.04; 3.32) | 0.42 |
|   Mossi | −4.59 | (−10.58; 1.41) | 0.13 | −4.11 | (−10.04;1.82) | 0.17 |
|   Peulh | −5.84 | (−12.02; 0.35) | 0.06 | −4.08 | (−10.22; 2.05) | 0.19 |
|   Semo | −1.98 | (−8.26; 4.30) | 0.54 | −1.77 | (−7.99; 4.44) | 0.58 |
| Body mass index | | | | 0.75 | (0.54; 0.95) | <0.001 |
| PHQ-9 | | | | 0.15 | (−0.08; 0.39) | 0.20 |
| Sitting hours per week | | | | 0.05 | (−0.01; 0.12) | 0.08 |

CSI-4, Couple Satisfaction Index-4; PHQ-9, Patient Health Questionnaire.

## Gender as a moderator

The interaction term (gender×CSI-4) was not statistically significant, neither in the linear regression model predicting SBP (B=0.03; 95% CI (−0.08 to 0.47); p=0.88), nor in the linear regression model predicting DBP (B=0.01; 95% CI (−0.10 to 0.24); p=0.92), reflecting that there was no evidence of gender differences in the association of relationship satisfaction with BP.

## DISCUSSION

The results of our population-based analysis of older adults in Burkina Faso suggest that relationship satisfaction is not associated with DBP. However, relationship satisfaction was positively associated with SBP, although the association was no longer significant after controlling for additional health-related parameters. The correlation coefficient between relationship satisfaction and SBP, although small, was comparable in magnitude to previously reported, mostly small effect sizes of the association between marital quality and health outcomes (r=0.07–0.21)[9] that have important implications at a population level. In our multiple regression model, adjusted for demographic and socioeconomic characteristics, relationship satisfaction was significantly and positively associated with an increase in SBP of 0.23 mmHg per unit change in the CSI-4. The effect size corresponds to roughly half of the SBP-increasing effect of 1 year of ageing and points in a direction that suggests, contrary to most prior findings,[9] that more relationally satisfied individuals in this population have higher SBP levels.

Given that the association of relationship satisfaction and SBP was no longer significant after controlling for additional health-related parameters, these findings should be interpreted with caution. BMI and depressive symptoms were significantly correlated with both relationship satisfaction and SBP and could act as confounders in the relationship between the two. In contrast to the negative associations of BMI on relationship satisfaction typically observed in Western populations, BMI demonstrated a positive correlation with relationship satisfaction in our sample, emphasising the importance of context-specific interpretations and the possibility of inverse associations between relationship satisfaction and a variety of health outcomes.[24] In Burkina Faso, cultural perceptions of body size may differ, potentially viewing higher BMI as a sign of attractiveness and wealth, which could positively influence relationship satisfaction.[24] Nevertheless, the observed associations of depressive symptoms, wealth and education with relationship satisfaction were consistent with patterns observed in Western cultural contexts: depressive symptoms were associated with lower relationship satisfaction, while wealth and education were

Table 6 Results of regression models estimating the association between relationship satisfaction and diastolic blood pressure, controlling for demographic and socioeconomic characteristics (model 1) and additionally relevant health-related variables (model 2)

| | Model 1 | | | Model 2 | | |
|---|---|---|---|---|---|---|
| | Estimate (B) | 95% CI | P value | Estimate (B) | 95% CI | P value |
| CSI-4 | 0.09 | (−0.04; 0.22) | 0.18 | 0.07 | (−0.06; 0.21) | 0.28 |
| Age (years) | 0.03 | (−0.02; 0.08) | 0.21 | 0.04 | (−0.01; 0.10) | 0.12 |
| Female (vs male) | −0.89 | (−1.87; 0.08) | 0.07 | −1.33 | (−2.30; −0.36) | 0.01 |
| Household wealth (vs poorest quintile) | | | | | | |
| Second poorest quintile | −0.57 | (−2.15; 1.00) | 0.47 | −0.77 | (−2.33; 0.79) | 0.34 |
| Middle quintile | 0.47 | (−1.11; 2.05) | 0.56 | −0.04 | (−1.61; 1.54) | 0.96 |
| Second richest quintile | 0.56 | (−1.03; 2.15) | 0.49 | 0.06 | (−1.52; 1.65) | 0.94 |
| Richest quintile | 1.94 | (0.27; 3.61) | 0.02 | 0.56 | (−1.14; 2.26) | 0.52 |
| Education (vs no education) | 1.93 | (0.05; 3.81) | 0.04 | 1.48 | (−0.39; 3.35) | 0.12 |
| Ethnicity (vs other ethnicities) | | | | | | |
| Bwaba | −2.15 | (−5.73; 1.43) | 0.24 | −1.87 | (−5.41; 1.68) | 0.30 |
| Dafin | −2.62 | (−6.17; 0.92) | 0.15 | −2.26 | (−5.77; 1.26) | 0.21 |
| Mossi | −2.94 | (−6.64; 0.76) | 0.12 | −2.65 | (−6.32; 1.02) | 0.16 |
| Peulh | −3.63 | (−7.45; 0.19) | 0.06 | −2.57 | (−6.37; 1.22) | 0.18 |
| Semo | −2.64 | (−6.52; 1.24) | 0.18 | −2.49 | (−6.33; 1.36) | 0.20 |
| Body mass index | | | | 0.44 | (0.31; 0.57) | <0.001 |
| PHQ-9 | | | | 0.06 | (−0.08; 0.21) | 0.40 |
| Sitting hours per week | | | | 0.03 | (−0.01; 0.06) | 0.17 |

CSI-4, Couple Satisfaction Index-4; PHQ-9, Patient Health Questionnaire.

associated with higher relationship satisfaction, consistent with the often-found association of mental health and socioeconomic stability with relationship quality.[52] Furthermore, the post hoc ANOVA and Tukey-HSD results examining the relationship between BMI and wealth showed that individuals in higher wealth quintiles had significantly higher BMI scores compared with those in lower quintiles. Given that wealthier individuals also report higher relationship satisfaction and considering the significant positive correlation between BMI and BP in our sample, these findings imply that the link between higher relationship satisfaction and increased BP may be partially mediated by BMI.

However, the potential association of relationship satisfaction with higher cardiovascular risk warrants further attention, as, in general, relationship satisfaction has been shown to be associated with health benefits. In contrast, the review by Kiecolt-Glaser and Wilson[23] synthesises empirical data demonstrating health risks associated with high-satisfaction partnerships, particularly in response to a partner's suffering, including elevated BP. In closer and more satisfying relationships, greater emotional interdependence can amplify negative emotions if one partner is suffering and encourage a more compassionate reaction, making one spouse's psychological and physiological issues more impactful on the other. Kiecolt-Glaser and Wilson[23] suggest that this effect is more likely to manifest among older adults with narrowing social networks and long-term shared experiences, resulting in increased vulnerability to these adverse health effects. The paradoxical findings of high relationship satisfaction being a risk factor for health particularly in distressing life situations and presumably more likely for older adults could partially explain the unexpected direction of our results. In Burkina Faso, the high prevalence of chronic diseases,[53] mental disorders and vital threats such as malnutrition and violence[30] creates an environment where the adverse effects of high relationship satisfaction could more readily contribute to negative health outcomes. Given that the family plays a central role in coping with stressors,[54] a high relationship satisfaction could exacerbate the negative health impact if the stress arises from spousal caregiving or spousal's suffering.[23 55]

While most studies on relationship satisfaction have focused on monogamous relationships, understanding the context of a polygamous setting is essential for interpreting the results. The complex dynamics inherent in polygamous marriages, a marital type likely common in our sample, may lead to a higher prevalence of ambivalent relationships. These relationships are characterised by both high positive (eg, social support) and high negative (eg, conflict) aspects and have been linked to cardiovascular risk factors such as greater 24-hour BP and greater coronary artery calcification.[56] According to

the review by Ross *et al*,[56] positive and negative aspects of close, ambivalent relationships interact in a way that positive aspects might not be health-promoting, while negative aspects could potentially pose even greater risks to health than in low-quality social ties. A high relationship satisfaction, presumably functioning as a positive aspect, could therefore contribute to adverse health effects in the context of this heterogeneous sample. Future research should address both positive and negative aspects of relationship quality separately, in order to obtain more information regarding the association with health outcomes.

### Gender differences

The association between relationship satisfaction and BP was not moderated by gender, which is consistent with Kurniawan *et al*,[24] who similarly found no gender differences through the utilisation of an interaction term analysis. This aligns with previous findings that found no gender differences in the relationship between marital satisfaction and mortality in a general American population sample that directly tested gender differences.[57] When separate analyses for gender were conducted, Robles *et al*,[9] on the other hand, reported statistically significant, larger associations between marital quality and surrogate cardiovascular endpoints in women than in men. It was discussed that the small, mostly insignificant gender differences were possibly due to too little power to measure small effect sizes or a smaller proportion of women in the sample. Both aspects, as well as a possible overestimation of relationship satisfaction for women (as explained further below), may have influenced our study's findings as well.

### Relationship satisfaction

This study represents one of the first investigations to examine relationship satisfaction in a Burkinabe population. The level of relationship satisfaction in our sample indicates that most participants experienced relationship distress, as about 64% (N=1345) of individuals reported CSI-4 scores below the cut-off value of 13.5. However, this is likely a slight overestimation of individuals scoring in the distressed range, given that the maximum possible score on the CSI-4 as implemented in the current study was one point lower compared with the original scale. If the cut-off was lowered by one point to 12.5, still over half of our sample, 53.7% (N=1136), would be considered distressed in their relationships. Women reported poorer overall relationship satisfaction than men, according to mean CSI-4 scores. This gender difference could be due to sociocultural specific characteristics, especially gender roles and patriarchy,[58] given the results of Jackson *et al*'s[59] meta-analysis indicating that there are no differences in marital satisfaction between men and women in community-based samples.

### Methodological considerations

Methodological considerations must also be taken into account when interpreting our findings. Robles *et al*[9]

noted a more consistent association between relationship satisfaction and structural markers of CVD compared with functional indicators like BP. Although studies using BP as a surrogate marker for cardiovascular risk have mostly found beneficial effects of high relationship satisfaction, the results have not always been uniform, nor has the context in which BP was considered—including BP measurements during interactions with spouses, as the mean from 24-hour (ambulatory BP) or from a one-time series of BP measurements (clinical BP).[9] One study, reporting findings in line with our results using a Western sample, is that of Cornelius *et al*,[19] which showed that higher relationship satisfaction was associated with higher 24-hour mean SBP, and not with DBP values. The authors suggest the effects of high relationship satisfaction possibly being time-dependent, reflecting both short-term reactive BP fluctuations and long-term cardiovascular adaptations that may influence outcomes in opposing ways.[19] The variability in BP-related outcomes results across studies may underscore the diverse pathways through which relationship satisfaction might influence cardiovascular health.

### Strengths and limitations

A significant strength of the current study is that it is large, population-based and representative of local adults from the age of 40 years onwards who are married or have a partner. Accordingly, our findings are locally valid and may also be transferable to other rural areas in Burkina Faso and other sub-Saharan African countries. Using both standardised questionnaires and physical measurements, internationally established health parameters could be objectively quantified and are thus comparable to other studies.

Nevertheless, several limitations in our study should be acknowledged. While the CSI-4 was highly reliable in our sample on a statistical level, its translation into the widely spoken local language during interviews makes it susceptible to differences in expression. Additionally, the unidimensional way to assess relationship satisfaction may not fully capture the positive and negative aspects present in the diverse range of relationships within a sample of both polygamous and monogamous unions, potentially limiting the nuanced and accurate representation of relationship satisfaction.[60] Furthermore, the study design did not account for marital type, wife-ordering, or inter-relations, despite the possibility that the sample includes non-cohabiting couples from 6 villages with fewer than 50 age-eligible participants.[45] Moreover, the household survey setting may have compromised privacy, potentially leading to an overestimation of relationship satisfaction in females, as women, whose social status is deeply connected to their husbands, often face social pressures to withhold personal feelings that could be interpreted as a challenge to the established social order.[33]

The current study used clinical BP as a surrogate marker for cardiovascular risk. Since the association of relationship satisfaction on health outcomes seems to

be situation and time-dependent, the use of 24-hour BP measurements in the home and everyday environment might be more accurate for determining BP values and may potentially lead to greater power in detecting the association between relationship satisfaction and BP.

Limitations that might have prevented the detection of associations between relationship satisfaction, DBP and SBP after controlling for health-related factors could stem from the study design: the use of clinical rather than ambulatory BP and a brief screening measure for relationship satisfaction rather than a longer unidimensional (eg, 16-item or 32-item CSI) or two-dimensional measure (distinct positive and negative dimensions), which might be able to capture more nuanced variations in relationship satisfaction. However, another reason could be that other predictors are in fact much stronger drivers for BP in the setting of Burkina Faso or because of possible confounding effects. Lastly, the cross-sectional nature of our study cannot demonstrate causation.

## CONCLUSIONS

Our study is the first to examine the direct association of relationship satisfaction and BP in rural sub-Saharan Africa using a study design that offers initial, valuable insights into the role of relationship satisfaction for cardiovascular risk factors in the context of low-income countries in sub-Saharan Africa, a sociodemographic and economic setting that has been largely ignored in this research field. Our data suggest that relationship satisfaction contributes marginally to explaining SBP variance compared with established BP drivers, especially age and BMI,[61] in Burkina Faso and interestingly in the opposite direction to what has mainly been described in other settings. These findings provide evidence for the importance of the contextuality of the relationship between relationship satisfaction and health. Further research is needed in the context of low-income sub-Saharan Africa to better understand the effects of relationship satisfaction in this sociocultural setting and to draw general conclusions about the pathways through which relationship satisfaction might exert influence. To examine the direct effect of relationship satisfaction in sub-Saharan Africa, future studies should address clinical outcomes such as mortality, as well as other cardiovascular surrogate endpoints such as ambulatory BP, and use measures of relationship satisfaction that can particularly capture the positive and negative aspects of relationships in diverse populations. Furthermore, the moderating role of relationship satisfaction on the stress-health link should be investigated further in the context of low-income countries, as it may yield interesting results regarding relevant sample characteristics such as high-prevalence chronic disease or family roles.

**Author affiliations**
¹Institute of Medical Psychology, Heidelberg University Hospital, Heidelberg, Germany
²Medical Faculty, Heidelberg University, Heidelberg, Germany
³Institute for Global Health, University College London, London, UK
⁴Africa Health Research Institute, KwaZulu-Natal, South Africa
⁵MRC/Wits Rural Public Health & Health Transitions Research Unit (Agincourt), University of the Witwatersrand, Johannesburg, South Africa
⁶School of Nursing & Public Health, University of KwaZulu-Natal, Durban, South Africa
⁷Centre de Recherche en Sante de Nouna, Nouna, Boucle du Mouhoun, Burkina Faso
⁸Institute of Global Health, University of Heidelberg, Heidelberg, Germany
⁹German Center for Mental Health (DZPG), partner site Heidelberg-Mannheim-Ulm, Germany
¹⁰Ruprecht Karls University, Heidelberg, Germany
¹¹Department of Psychology, Philipps-Universität Marburg, Marburg, Germany

**Acknowledgements** We would like to express our gratitude to all the interviewers and data collectors responsible for collecting and executing the field study as well as to the participants for taking part in the survey. Additionally, we would like to thank the attendees of the 37th Annual Conference of the European Health Psychology Society (EHPS) for their interest in our research and the valuable discussions during the poster session. The presented results have already been showcased at the conference, and the feedback received has contributed to the development of this manuscript. We refer to the published abstract by Jaspert et al (62), titled 'Association of relationship satisfaction and blood pressure in rural Burkina Faso's elderly population', presented at the EHPS in 2023, for further details.

**Contributors** TB, AS and GH conceived and designed the overall CRSN Heidelberg Aging Study. GH and AS coordinated data collection and preparation. MB contributed to data acquisition as coordinator of field data collection. FMJ conducted the analysis and wrote the manuscript. MSF, BD and TB supervised the analysis and manuscript development. All authors substantively reviewed manuscripts, inputted into revisions and approved the final manuscript.

**Funding** This work was supported by the Alexander von Humboldt Foundation through an Alexander von Humboldt Professorship award to TB, which is funded by the German Federal Ministry of Education and Research. GH is supported by a fellowship from the Wellcome Trust and Royal Society (grant number 210479/Z/18/Z). MSF was supported by a grant from the Baden-Wuerttemberg Ministry of Science, Research and the Arts (Germany) and the European Social Fund. This research was funded in whole, or in part, by the Wellcome Trust (grant number 210479/Z/18/Z). For the purpose of open access, the author has applied a CC BY public copyright licence to any Author Accepted Manuscript version arising from this submission.

**Disclaimer** The funders had no role in study design, data collection and analysis, decision to publish, or preparation of the manuscript.

**Competing interests** None declared.

**Patient and public involvement** Patients and/or the public were not involved in the design, or conduct, or reporting, or dissemination plans of this research.

**Patient consent for publication** Not applicable.

**Ethics approval** Ethical approval for this study was granted by the Ethics Commission of the Medical Faculty Heidelberg (S-120/2018), the Burkina Faso Comité d'Ethique pour la Recherche en Santé (CERS) in Ouagadougou (2018-4-045) and the Institutional Ethics Committee (CIE) of the CRSN (2018-04). Prior to participation, oral assent was obtained from all village elders. Additionally, written informed consent was obtained from each participant, and in cases of illiteracy, a literate witness assisted in the consent process.

**Provenance and peer review** Not commissioned; externally peer reviewed.

**Data availability statement** Data are available on reasonable request. Data may be obtained from a third party and are not publicly available. Data used in this study are not publicly available due to restrictions on data sharing imposed by the consent agreements with participants. The dataset includes entire age cohorts from specific villages, and there is a potential risk of deductive disclosure with sufficient local information. Anonymised data can be obtained from the CHAS study data controllers upon signing a data use agreement that restricts onward transmission.

Researchers interested in replicating the analyses or pursuing collaborative studies using CHAS data are encouraged to contact GH (g.harling@ucl.ac.uk) for further information and to discuss potential collaborations.

**Author note** In preparing this manuscript, ChatGPT was used to assist with refining text for clarity and accuracy, and DeepL was used for translation accuracy; all AI-generated content was reviewed and edited by the authors to ensure accuracy and originality. In accordance with BMJ Open's guidelines, FMJ and MSF serve as guarantors for this study, accepting full responsibility for the content and conduct of the research.

**ORCID iDs**
Felicitas Maria Jaspert http://orcid.org/0009-0005-2301-1516
Guy Harling http://orcid.org/0000-0001-6604-491X

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
