## [Reviewer comments · BMJ Open]

ARTICLE DETAILS

Title (Provisional)

The Association of Relationship Satisfaction with Blood Pressure: A Cross-Sectional Study of Older Adults in Rural Burkina Faso

Authors

Jaspert, Felicitas Maria; Harling, Guy; Sie, Ali; Bountogo, Mamadou; Bärnighausen, Till; Ditzen, Beate; Fischer, Melanie Sandy

VERSION 1 - REVIEW

Reviewer	1
Name	Trillingsgaard, Tea
Affiliation	Aarhus University Department of Psychology and Behavioural Sciences, Psychology
Date	05-Jul-2024
COI	None

This review was conducted in collaboration by Lea Tangelev, Anne Klode, Nanna Fensman Lassen and Tea Trillingsgaard.

This study examines the association between relationship satisfaction and blood pressure in 2,114 older adults, representative of the area in rural Burkina Faso. This research is novel and interesting due to its unique cultural and economic context, which is rarely seen in studies on social relationships. As highlighted by the authors, the link between relationship satisfaction and health is sensitive to context, and the results confirm this by some results that would be unexpected in Western-world context. Overall, the manuscript is very well-written with a clear presentation of methods, results, and conclusions. However, the context of the study as well as previous findings in the field could be more clearly framed in the introduction, a correlation matrix is missing from results, and the interpretation of findings is difficult to follow. Below are our recommendations to the authors.

Introduction:

- It is necessary that the number of abbreviations is reduced to improve the readability of this manuscript.
- On page 7, lines 42-49, please indicate that reference 9 is a meta-analysis.
- A theoretical explanation for the link between relationship satisfaction and blood pressure/cardiovascular disease should be presented in the introduction.

- We learn in the method section that 42 % of married women live in polygamous marital relationships. This context should be clearly framed in the introduction, as this sets the stage for understanding and measuring satisfaction in a couple relationship. How is the constitution of marital and couple relationships different or similar to Western countries? How are families living together and what are the options for divorce? Are marriages arranged?
- A review of previous studies examining the relationship between relationship satisfaction and blood pressure/cardiovascular disease in Western Countries is missing from the introduction.
- Page 7, lines 22-24: “while data on the influence of RS on BP remain scarce or even non-existent”. It is unclear whether this sentence refers to SSA specifically or research in general.

Methods:

- A correlation matrix of all included variables is missing from the tables. This is needed to understand potential confounders, variables linked to both relationship satisfaction and blood pressure, in particular the level of wealth and BMI.
- The measure and labels of education are somewhat confusing. Please clarify the coding, 0 = did not complete primary school, 1= completed primary school or more. Define “primary school” and describe how this information was collected. If more fine-grained levels of education are available in the dataset, these would be relevant.
- Table 1. Be aware of the identifiable data in cells with just one case (e.g. 1 widow), consider using masking (e.g. widows < 3)
- Among the 2,114 participants, were any of them couples? How many were living in polygamy? If so, were wives first or second wife? Were they interviewed together with more family members or in private? Were interviews conducted in the presence of an interpreter?

Discussion:

- Among the many potential contextual factors that could explain these unexpected findings, the authors highlight that family members have high caregiving strain, which may be linked to both high relationship satisfaction and blood pressure. This interpretation is hard to follow. Can the authors provide support that higher caregiving stress is associated with higher relationship satisfaction?
- The authors discuss the limitation of using clinical BP as opposed to ambulatory BP, the latter being more sensitive. Although this limitation is relevant to mention, it cannot explain the positive association between relationship satisfaction and BP, and so we still need that interpretation.
- It is important to discuss other aspects of the context in the interpretation of both relationship satisfaction levels and levels of blood pressure.
- The finding that the association between relationship satisfaction and blood pressure is no longer significant when health related variables are accounted for needs some interpretation. Specifically, we would like to see a discussion of health and wealth, and any potential confounders as they may show in the correlation matrix requested above. Understanding links between BMI, health, wealth, and relationship satisfaction in this specific context may help interpret results.

Reviewer	2
Name	Ren, Yongcheng
Affiliation	Huanghai University
Date	05-Aug-2024
COI	no.

Appropriate methods, provide effective evidence for the predictive evidence for Blood Pressure in a Population-Based Sample of Older Adults in Rural Burkina Faso. We would like to pose some clarifying questions and suggestions to the authors.

1. In methods, the main concern regarding the models is covariates selected for the adjustment. What was the logic for the selection of these variables as confounders?

2. The author analyzed systolic and diastolic blood pressure as continuous variables. I wonder if the impact of antihypertensive medications on blood pressure values was taken into consideration?

3. Suggest citing articles (DOI=10.1038/s41598-024-60906-w) in the Discussion or Introduction section to increase its breadth, as it delved into examining the correlation between other composite indices and anomalies in blood press levels, offering insights into the multifaceted relationship between metabolic markers and blood press dysregulation.

VERSION 1 - AUTHOR RESPONSE

Reviewer 1: Dr. Tea Trillingsgaard

This review was conducted in collaboration by Lea Tangelev, Anne Klode, Nanna Fensman Lassen and Tea Trillingsgaard.

This study examines the association between relationship satisfaction and blood pressure in 2,114 older adults, representative of the area in rural Burkina Faso. This research is novel and interesting due to its unique cultural and economic context, which is rarely seen in studies on social relationships. As highlighted by the authors, the link between relationship satisfaction and health is sensitive to context, and the results confirm this by some results that would be unexpected in Western-world context. Overall, the manuscript is very well-written with a clear presentation of methods, results, and conclusions. However, the context of the study as well as previous findings in the field could be more clearly framed in the introduction, a correlation matrix is missing from results, and the interpretation of findings is difficult to follow. Below are our recommendations to the authors.

We appreciate the reviewers' positive feedback on our study design and manuscript presentation, as well as their recognition of the unique cultural and economic context of our research in the field of social relationships. In the following, we are pleased to address our adjustments to the manuscript based on the reviewers' comments.

1. Introduction

- It is necessary that the number of abbreviations is reduced to improve the readability of this manuscript.**

We have reduced the number of abbreviations to improve readability, particularly abbreviations such as DALY, NCD, SSA and CHAS.

- **On page 7, lines 42-49, please indicate that reference 9 is a meta-analysis.**

We have clarified that reference 9 is a meta-analysis by modifying the text to: “Although married people generally exhibit better health than unmarried people, as demonstrated in a meta-analysis by Robles *et al.* (9)...” (p. 7, ll. 107-110).

- **A theoretical explanation for the link between relationship satisfaction and blood pressure/cardiovascular disease should be presented in the introduction.**

We thank the reviewer for this suggestion and agree that this point should have been clearer. We have expanded the introduction to include a theoretical explanation of the link between relationship satisfaction and cardiovascular health discussing the reactivity hypothesis as a central framework. This theory posits that relationship satisfaction influences cardiovascular health through its impact on stress reactivity, where positive relationships buffer against stress-induced cardiovascular responses, while negative interactions exacerbate them. We also address the complex role of stress transmission within close relationships, particularly in high-quality marriages where one partner's health issues can negatively affect the other's cardiovascular health (pp. 7-8, ll. 128-140).

- **We learn in the method section that 42 % of married women live in polygamous marital relationships. This context should be clearly framed in the introduction, as this sets the stage for understanding and measuring satisfaction in a couple relationship. How is the constitution of marital and couple relationships different or similar to Western countries? How are families living together and what are the options for divorce? Are marriages arranged?**

We appreciate this point and have revised the introduction accordingly. We have framed the study within the cultural context of Burkina Faso, highlighting the distinct nature of marriage practices, including the prevalence of polygamous unions, arranged marriages, and the differing marital expectations compared to Western norms. This context is crucial for understanding how relationship satisfaction might be conceptualized in Burkina Faso, where marital roles often emphasize family obligations and social stability over personal emotional fulfillment. These cultural factors are now clearly outlined to provide a comprehensive background for interpreting our findings (pp. 9-10, ll. 178-194).

- **A review of previous studies examining the relationship between relationship satisfaction and blood pressure/cardiovascular disease in Western Countries is missing from the introduction.**

We have added a brief summary of previous studies from Western countries, highlighting the significant association between high relationship satisfaction and better cardiovascular health outcomes (p 7, ll. 126-128).

- **Page 7, lines 22-24: “while data on the influence of RS on BP remain scarce or even non-existent”. It is unclear whether this sentence refers to SSA specifically or research in general.**

We thank the reviewer for pointing this out. We clarified that the statement on the scarcity of data on the influence of RS on BP refers specifically to the sub-Saharan region (SSA) (pp. 9-8, ll. 154-157).

2. Methods

- **A correlation matrix of all included variables is missing from the tables. This is needed to understand potential confounders, variables linked to both relationship satisfaction and blood pressure, in particular the level of wealth and BMI.**

To account for potential confounding effects of covariates that might influence both RS and BP values, Pearson's correlation coefficients were calculated for all pairs of continuous variables to assess linear relationships (Table 3, p. 18). For categorical

variables, mean RS and BP values were computed for each category (Table 4, p. 19). Differences between binary categories were assessed using t-tests, while differences across multiple categories were evaluated using one-way ANOVA, providing insight into the statistical significance of group differences.

- **The measure and labels of education are somewhat confusing. Please clarify the coding, 0 = did not complete primary school, 1= completed primary school or more. Define “primary school” and describe how this information was collected. If more fine-grained levels of education are available in the dataset, these would be relevant.**

We agree with the reviewer that these categories require additional clarification. The binary coding of educational attainment into "did not complete primary school" versus "completed primary school or more" was driven by the highly skewed distribution of education in the sample (83% of participants had no formal schooling, 10% less than primary education, and only 7% completed primary school or higher), the context-specific significance of primary school completion, and the need for clear, interpretable results. While finer-grained educational levels are available, cell sizes were extremely small. Therefore, they were combined to avoid issues with statistical power and to maintain focus on a key educational threshold that is highly relevant in the rural Burkina context.

We clarified the coding of education and defined primary school as well as its relevance in the context of Burkina Faso (p. 14, ll. 280-288).

- **Table 1. Be aware of the identifiable data in cells with just one case (e.g. 1 widow), consider using masking (e.g. widows < 3)**

We appreciate the reviewer’s attention to the potential for individual data points to be identifiable. In order to avoid confusion regarding percentage points not matching the numbers (or percentage points indirectly pointing to single cases), we chose to combine the marital statuses of separated, divorced, and widowed into a single category (combined < 1% of the sample) to protect participant confidentiality (Table 1, p. 16).

- **Among the 2,114 participants, were any of them couples? How many were living in polygamy? If so, were wives first or second wife? Were they interviewed together with more family members or in private? Were interviews conducted in the presence of an interpreter?**

We clarified that interviews were conducted individually, with the setting (public or private) chosen by the interviewees. Local fieldworkers conducted the interviews in the local languages without the need for interpreters (p. 12, ll.233-237). We have added as limitations that the household survey setting may have compromised privacy, potentially leading to an overestimation of relationship satisfaction among females due to social pressures, and that the study did not account for marital type, wife-ordering, or inter-relations in a sample that may include couples (p. 27, ll. 468-474).

3. Discussion

- **Among the many potential contextual factors that could explain these unexpected findings, the authors highlight that family members have high caregiving strain, which may be linked to both high relationship satisfaction and blood pressure. This interpretation is hard to follow. Can the authors provide support that higher caregiving stress is associated with higher relationship satisfaction?**

We expanded the discussion on the link between caregiving strain, RS, and BP. We included findings from the review by Kiecolt-Glaser and Wilson on convergence-related health risks in high-satisfaction partnerships and Monin et al.'s hypothesis that high RS may amplify the negative health impact of spousal caregiving (pp. 23-24, ll. 386-401).

- **The authors discuss the limitation of using clinical BP as opposed to ambulatory BP, the latter being more sensitive. Although this limitation is relevant to mention, it cannot explain the positive association between relationship satisfaction and BP, and so we still need that interpretation.**

We agree that the discussion of the psychological and physiological factors that may explain the positive association is most central and, as noted above have expanded our discussion to address this point, as well as the possibility of adverse health effects in ambivalent relationships, which may be particularly relevant in our study's context (pp. 24-25, ll. 402-414). At the same time, we also believe that the assessment of slightly different blood-pressure related endpoints are important to consider and have attempted to clarify this point in the paragraph on methodological considerations, referencing relevant studies and highlighting the potential for multiple, context-dependent pathways through which RS may influence BP (p. 26, ll. 440-454).

- **It is important to discuss other aspects of the context in the interpretation of both relationship satisfaction levels and levels of blood pressure.**

We agree that incorporating contextual factors into the interpretation of relationship satisfaction and blood pressure is crucial. We have expanded our discussion to include the dynamics of polygamous relationships and their effects on health and marital functioning (p. 10, ll. 181-187). Additionally, we explored the potential adverse health outcomes, such as elevated blood pressure, associated with ambivalent relationships, where both high positive and high negative aspects interact (pp. 24-25, ll. 402-414).

- **The finding that the association between relationship satisfaction and blood pressure is no longer significant when health related variables are accounted for needs some interpretation. Specifically, we would like to see a discussion of health and wealth, and any potential confounders as they may show in the correlation matrix requested above. Understanding links between BMI, health, wealth, and relationship satisfaction in this specific context may help interpret results.**

We added a correlation matrix for continuous variables, mean RS and BP values for each category of categorical variables and a discussion on possible confounding effects, where we discussed how BMI, depressive symptoms, wealth, and education may act as potential confounders in the relationship between RS and BP. We emphasized the importance of context-specific interpretations (p. 23, ll. 372-384). Furthermore, we discussed potential limitations that might have prevented the detection of associations between RS, DBP, and SBP after controlling for health-related factors in the strengths and limitations section (p. 27, ll. 480-487).

Reviewer 2: Dr. Yongcheng Ren

Appropriate methods, provide effective evidence for the predictive evidence for Blood Pressure in a Population-Based Sample of Older Adults in Rural Burkina Faso. We would like to pose some clarifying questions and suggestions to the authors.

We thank the reviewer for their positive feedback on our methods and the effectiveness of our evidence for predicting blood pressure in a population-based sample of older adults in rural Burkina Faso. We appreciate your thoughtful questions and suggestions, and we are eager to address them to further clarify and enhance our study.

- **In methods, the main concern regarding the models is covariates selected for the adjustment. What was the logic for the selection of these variables as confounders?**

We acknowledge that the rationale for selecting covariates for adjustment may not

have been thoroughly articulated in the methods section; therefore, we have included a brief clarification to address this (p. 13, ll. 274-276). Variables were chosen based on their established relevance in epidemiological studies and their known influence on RS and BP. Specifically, variables such as age, BMI, physical activity, and depressive symptoms are well-documented factors that can impact both BP and RS, thus potentially confounding the relationship between these two variables. Our goal was to control for these confounders to ensure a more accurate assessment of the association between RS and BP in our study population.

- **The author analyzed systolic and diastolic blood pressure as continuous variables. I wonder if the impact of antihypertensive medications on blood pressure values was taken into consideration?**

We agree with the reviewer that potential effects of antihypertensive medications are important to consider. We addressed this issue by excluding participants who reported taking antihypertensive medications within the last two weeks prior to the study. This exclusion is noted at the beginning of the results section, where we specify that 101 participants were not included in the analyses due to recent antihypertensive medication use (p. 15, ll. 319-320). By excluding these individuals, we aimed to mitigate the potential confounding effects of medication on BP values, ensuring that our results reflect the relationship between RS and BP independent of pharmaceutical intervention.

- **Suggest citing articles (DOI=10.1038/s41598-024-60906-w) in the Discussion or Introduction section to increase its breadth, as it delved into examining the correlation between other composite indices and anomalies in blood press levels, offering insights into the multifaceted relationship between metabolic markers and blood press dysregulation.**

We appreciate the recommendation to include the article by Dr. Yongcheng Ren *et al.* (DOI: 10.1038/s41598-024-60906-w). We have incorporated the reference into the Introduction to highlight the emerging research on composite indices that distinguish cardiovascular health levels and their correlation with high-normal blood pressure in older adults. This addition emphasizes the importance of understanding these associations, particularly as relationship satisfaction is a potentially modifiable risk factor, which could inform targeted interventions such as relationship education programs (p. 9, ll. 157-161).

We hope that these revisions satisfactorily address all the comments and suggestions provided by the editor and reviewers. We believe these changes have strengthened our manuscript and provided a clearer, more comprehensive presentation of our research findings.

Thank you again for your time and consideration. We look forward to your feedback.

Sincerely,
Felicitas M. Jaspert

VERSION 2 - REVIEW

Reviewer	1
Name	Trillingsgaard, Tea
Affiliation	Aarhus University Department of Psychology and Behavioural Sciences, Psychology

Date 19-Sep-2024

COI

The authors' response to the review was thorough and the manuscript has substantially improved. Some minor concerns are listed below.

Abbreviations. The readability of the manuscript has been significantly improved by fewer abbreviations. However, we encourage the authors to also refrain from abbreviating 'relationship satisfaction.' Additionally, they should be mindful of the correct and consistent use of abbreviations, including introducing abbreviations upon the first use of the term (for example, BP line 127 + 133 and diastolic BP line 201 + 236). We refer to the APA manual for the correct APA usage of abbreviations.

Theory. The authors have provided a theoretical explanation for the link between relationship satisfaction and BP/CVD. However, the claim that high relationship quality can have detrimental effects on health, is not warranted ("Particularly in high-quality relationships, stress can be transmitted between partners if one partner has health problems which emphasizes the possible detrimental effects of high relationship quality on health", line 138-140). We suggest that the authors refine the phrasing and clarify that it is the possibility for spillover from shared burdens in relationships that may contribute to worse health and account for the association between relationship satisfaction and decreased health.

Previous findings. The authors have provided a few sentences on the existing research ("In Africa, very little research exists specifically addressing RS or its impact (e.g. 15-17). Previous research in Western countries demonstrate a significant association between high RS and better cardiovascular health, including lower BP, reduced risk of coronary heart disease, improved heart rate variability, as well as decreased incidence of myocardial infarction (9,18,19)", line 125-128). What is the nature of previous evidence for the link between RS and BP, is it extensive, consistent, a solid effect size ...or is it sparse, mixed and inconsistent? This information would improve the interpretation of how the current findings in light of the existing evidence-base.

Analyses. The authors have provided a correlation matrix as well as t-tests and ANOVA results. However, from these analyses, we still cannot see whether the positive association between relationship satisfaction and higher BP could be explained by a BMI-wealth link, as wealth is not included in the correlation matrix. If a continuous measure is available from the wealth index, the authors should include this in the matrix.

Reviewer 2
Name Ren, Yongcheng

Affiliation	Huanghai University
Date	24-Sep-2024
COI	no

no comments.

VERSION 2 - AUTHOR RESPONSE

Reviewer 1: Dr. Tea Trillingsgaard

The authors' response to the review was thorough and the manuscript has substantially improved. Some minor concerns are listed below.

Thank you for your positive feedback. We are pleased that the revisions have substantially improved the manuscript and have thoughtfully considered the remaining minor concerns. Below, we provide a detailed response to each point.

- 1. Abbreviations. The readability of the manuscript has been significantly improved by fewer abbreviations. However, we encourage the authors to also refrain from abbreviating 'relationship satisfaction.' Additionally, they should be mindful of the correct and consistent use of abbreviations, including introducing abbreviations upon the first use of the term (for example, BP line 127 + 133 and diastolic BP line 201 + 236). We refer to the APA manual for the correct APA usage of abbreviations.**

We appreciate your careful attention regarding the use of abbreviations. We have removed the abbreviation for 'relationship satisfaction' throughout the manuscript to further enhance readability. Additionally, we have reviewed the manuscript to ensure the correct and consistent use of abbreviations as per APA guidelines, including introducing them at first mention as well as defining all abbreviations used in tables in the corresponding table note.

- 2. Theory. The authors have provided a theoretical explanation for the link between relationship satisfaction and BP/CVD. However, the claim that high relationship quality can have detrimental effects on health, is not warranted (“Particularly in high-quality relationships, stress can be transmitted between partners if one partner has health problems which emphasizes the possible detrimental effects of high relationship quality on health”, line 138-140). We suggest that the authors**

refine the phrasing and clarify that it is the possibility for spillover from shared burdens in relationships that may contribute to worse health and account for the association between relationship satisfaction and decreased health.

We agree with the reviewer that the phrasing needs refinement. We have revised the theoretical explanation to more accurately reflect that it is not the high quality of relationships per se that has detrimental effects on health, but rather the potential for stress-spillover effects between partners in high-quality relationships that can influence the link between relationship quality and health. The relevant section has been updated to better align with this refined understanding (p. 8, ll. 142-144).

- 3. Previous findings. The authors have provided a few sentences on the existing research (“In Africa, very little research exists specifically addressing RS or its impact (e.g. 15-17). Previous research in Western countries demonstrate a significant association between high RS and better cardiovascular health, including lower BP, reduced risk of coronary heart disease, improved heart rate variability, as well as decreased incidence of myocardial infarction (9,18,19)”, line 125 -128). What is the nature of previous evidence for the link between RS and BP, is it existensive, consistent, a solid effect size ...or is it sparse, mixed and inconsistent? This information would improve the interpretation of how the current findings in light of the existing evidence-base.**

Thank you for your valuable comment. We have revised the manuscript to better reflect the current state of evidence. We now highlight that research regarding the association between relationship satisfaction and blood pressure specifically is limited, with studies generally reporting small positive effect sizes, along significant heterogeneity across studies (p. 8, ll. 129-132). This acknowledges that findings on the relationship between relationship satisfaction and BP are not entirely consistent, which we elaborate more on in the discussion section addressing methodological considerations.

- 4. Analyses. The authors have provided a correlation matrix as well as t-tests and ANOVA results. However, from these analyses, we still cannot see whether the positive association between relationship satisfaction and higher BP could be explained by a BMI-wealth link, as wealth is not included in the correlation matrix. If a continuous measure is available from the wealth index, the authors should include this in the matrix.**

We appreciate the thoughtful feedback and agree that a BMI-wealth link is still missing in our analysis and interpretation of results. We would like to clarify that in our dataset, wealth is not available as a continuous variable, which is why it was not included in the initial correlation matrix.

As we value the addition of an investigation of the BMI-wealth link and its possible impact on the association between relationship satisfaction and BP, we performed a post-hoc ANOVA comparing BMI across wealth quintiles, followed by a Tukey-HSD test to identify specific differences between wealth quintiles with the results added as supplementary material. This analysis revealed that individuals in higher wealth quintiles had significantly higher BMI scores compared to those in lower quintiles. Given the previous reported results, these findings imply that the link between higher relationship satisfaction and increased BP may be partially mediated by BMI. We have updated the Results (p. 19, ll. 344-351) and Discussion (p. 26, ll. 410-415)

sections of our manuscript to reflect these insights and provide a more comprehensive understanding of the interplay between wealth, BMI, relationship satisfaction, and BP. We appreciate your guidance in refining our analysis and enhancing the manuscript.

Reviewer 2: Dr. Yongcheng

Ren No comments.

We hope that these revisions satisfactorily address all remaining comments and suggestions provided by the editor and reviewer. We believe these changes have further strengthened our manuscript and provided a clearer, more comprehensive presentation of our research findings.

Thank you again for your time and consideration. We look forward to your feedback.

Sincerely,
Felicitas M. Jaspert